# Modelling the Impact of Water Stress during Post-Veraison on Berry Quality of Table Grapes

Abdelmalek Temnani [1], Pablo Berríos [1], María R. Conesa [2] and Alejandro Pérez-Pastor [2,*]

1 Departamento de Ingeniería Agronómica, Universidad Politécnica de Cartagena (UPCT), Paseo Alfonso XIII, 48, ETSIA, 30203 Cartagena, Spain; abdelmalek.temnani@edu.upct.es (A.T.); pablo.berrios@edu.upct.es (P.B.)
2 Departamento de Riego, CEBAS-CSIC, P.O. Box 164, 30100 Murcia, Spain; mrconesa@cebas.csic.es
* Correspondence: alex.perez-pastor@upct.es; Tel.: +34-968-327-035

**Abstract:** The aims of this work were modelling the effect of water stress intensity during post-veraison on table grape quality and yield, as well as predicting berry quality at harvest using a machine learning algorithm. The dataset was obtained by applying different irrigation regimes in two commercial table grape vineyards during seven growing seasons. From these data, it was possible to train and validate the predictive models over a wide range of values for the independent (water stress intensity and fruit load) and dependent (firmness and berry color) variables. The supervised learning algorithm Gaussian Process Regression allowed us to predict the variables with high accuracy. It was also determined that a reduction in irrigation of up to 40% during post-veraison, compared to vines without water limitations, and the accumulation of the water stress integral of up to 30 MPa per day, linearly increase the irrigation water use efficiency (IWUE) and promote higher berry color and firmness. The severe water scarcity and the increasing uncertainty about the irrigation water availability for the season that farmers are facing highlight the advantage of incorporating these validated techniques into agricultural decision making, as they allow for the planning of cultural practices and criteria to increase the IWUE and crop sustainability.

**Keywords:** Gaussian Process Regression; radial basis function kernel; fruit load; berry color; berry firmness; water stress integral; CIRG





## 1. Introduction

Table grape (*Vitis vinifera* L.) production in Spain for the fresh market covers an area of 14,665 ha, of which around 92% is produced under irrigation and reaches an average yield of 24.2 t per ha. The Region of Murcia, in Southeastern Spain, is the main producing area and represents almost 50% of the national area [1]. Its production has been stimulated by the high-quality production and the market acceptance of seedless varieties such as 'Crimson Seedless' [2]. This cultivar was developed by the USDA (Fresno, CA, USA) and is characterized by late-harvesting, medium-sized, cylindrical, bright red berries with thick skin [3] and excellent organoleptic properties, such as a crunchy texture and a sweet taste [4,5]. Clusters are generally large, conical, and compact. It has high bud fertility and is therefore very productive. Vines grow on a wide range of soil types and can exhibit vigorous vegetative growth, which affects productivity and berry quality [6,7]. Likewise, the main problem associated with this cultivar is insufficient berry coloring [8–10], which, in combination with berry firmness, is the key factor determining consumer acceptance [11]. In Mediterranean climates, such as in the Region of Murcia, with high summer temperatures that inhibit the accumulation of anthocyanins [12,13] and a narrow temperature range between day and night during ripening, clusters with heterogeneous color can be observed, with fully red and still green berries [14,15], causing important economic losses. In this sense, different cultural practices have been investigated in 'Crimson Seedless' to avoid these problems: (i) the application of plant growth regulators such as abscisic acid and

Ethephon during berry growth, but these have shown inconsistent results [10,14–18]; (ii) canopy management, either by regulating vine vigor or thinning leaves close to the clusters to increase light exposure [7,9]; (iii) fruit load regulation, such as flower thinning, removal of set berries, or cluster thinning; this management strategy also has the advantage that it can be carried out at the beginning of the season until the berries have reached a size of around 5 mm [7,9,10]; (iv) the application of a deficit irrigation regime, which has appeared as a more sustainable alternative to prevent berry coloring problems and also promote the production of bioactive compounds [4,19,20].

Deficit irrigation (DI) can increase berry color and cluster homogeneity at harvest, and, at the same time, increase the irrigation water use efficiency (IWUE) without negatively affecting yield or berry quality [4,12,19,21]. The effect of climate change has increased the intensity of water scarcity in Mediterranean areas, so different DI strategies have been studied instead [22,23]. The most common methods are regulated deficit irrigation (RDI) [24] and partial root-zone drying (PRD) [25]. Both provide less irrigation during periods of the crop that are not sensitive to water deficit. In 'Crimson Seedless', the non-critical period is during post-veraison [12,21]. Another DI method that can increase the color of grape berries is sustained deficit irrigation (SDI), although, in contrast to RDI, the irrigation reduction is applied during the entire crop cycle [26–28]. However, the effect of SDI would reduce crop yield and vegetative growth in the long term [29,30]. Generally, the reduction in irrigation is estimated from the FAO water balance [31], but it is necessary to complement it with a method to control the plant water status [32]. In this sense, the most validated plant water status indicator is the stem water potential ($\Psi_s$), as it is directly related to environmental conditions and soil water availability [33–38]. Furthermore, the water stress integral [39] is an appropriate tool for quantifying the water stress applied as a function of the $\Psi_s$, and to extrapolate the protocols obtained to other agro-climatic zones. Therefore, determining the magnitude of the optimal crop water stress is essential to increase the sustainability of production.

The irrigation volume available to the farmer is increasingly uncertain, so it is necessary to explore the relationships between the intensity of the water stress to be applied and its effect on berry quality, as well as its interaction with cultural practices, to optimize the production. The incorporation of machine learning algorithms has made it possible to obtain models for several uses in agriculture [40,41]. Thus, using a Gaussian Process Regression [42], the weekly water requirement of 'Crimson Seedless' vines could be estimated with high accuracy from the daily maximum temperature and day of the year [43]. To obtain robust models, it is necessary to have a reliable database, obtained under controlled experimental conditions and with a wide range of values to train and validate the model in different scenarios. With this premise, our research was carried out using data obtained during several seasons from the research of our team in two experimental sites and with vines subjected to a wide range of water stress.

Therefore, our research aims to (i) model the effect of water stress intensity on berry quality and yield, and (ii) predict berry quality at harvest in a Mediterranean climate with severe water scarcity, based on two easily applicable and quantifiable parameters: water stress integral and fruit load. Both objectives were developed to provide farmers with a tool to cope with water scarcity while maintaining the production sustainability and increasing the irrigation water use efficiency.

## 2. Materials and Methods

### 2.1. Data Collection

The data were collected between 2011 and 2017, from two commercial table grape (*Vitis vinifera* L.) cv. Crimson Seedless vineyards (named *ES1* and *ES2* in the present study), located in Southeastern Spain. In these sites, different studies were carried out on this crop by Conesa et al. and Temnani et al. [19,21,43]. The reference crop evapotranspiration ($ET_0$) was obtained by the weather stations of the "Servicio de Información Agraria de Murcia" [44]. Data were computed as an average of the 7 previous days. The crop evapo-

transpiration ($ET_c$) was calculated according to the FAO method ($ET_c = ET_0 \times k_c$) [31], with the crop coefficients ($k_c$) reported by Williams and Ayars [45], varying between 0.2 and 0.8 according to the phenological stage. The dataset was obtained using different irrigation strategies, as follows. (i) Control (CTL): vines were irrigated at 110% of the $ET_c$ to avoid water restrictions throughout the irrigation season (from April to October). (ii) Regulated Deficit Irrigation (RDI): vines were irrigated without water restrictions, except during the non-critical period of post-veraison [21,43], when the vines were irrigated at 50% of CTL. (iii) Partial Root-zone Drying (PRD): vines were irrigated as with RDI but alternating the wet and dry sides of the root zone every 10–14 days [46–48], when 75% of the soil field capacity was reached in the dry root zone. (iv) Sustained Deficit Irrigation (SDI): vines irrigated at 50% of CTL throughout the entire irrigation season. (v) Null Irrigation (NI): vines received only rainwater and occasional supplementary irrigation when the stem water potential ($\Psi_s$) was below the threshold of −1.2 MPa previously determined for 'Crimson Seedless' [49]. The description of the experimental sites is detailed in Table 1.

**Table 1.** Description of table grape cv. Crimson Seedless experimental sites for data acquisition [19,21,43].

| | *Experimental Site 1 (ES1)* | *Experimental Site 2 (ES2)* |
|---|---|---|
| *Location* | Cieza, Murcia, Spain 38°15′0.09″ N, 1°32′60.00″ W | Molina de Segura, Murcia, Spain 38°6′52.14″ N, 1°10′29.36″ W |
| *Rootstock* | 1103-Paulsen | 1103-Paulsen |
| *Planting frame* | 4.0 × 4.0 m; 625 vines ha$^{-1}$ | 3.0 × 3.5 m; 952 vines ha$^{-1}$ |
| *Planting year* | 2001 | 2003 |
| *Irrigation system* | 4 drippers of 4 L h$^{-1}$ per vine One drip line per vine row | 3 drippers of 4 L h$^{-1}$ per vine One drip line (CTL, RDI, SDI, and NI) and two drip lines (PRD) per vine row |
| *Soil characteristics* | Soil texture class: clay–silt–loam Bulk density: 1.25 g cm$^{-3}$ Organic matter: 2.1% Soil pH: 8.6 FC: 0.34 m$^3$ m$^{-3}$ WP: 0.18 m$^3$ m$^{-3}$ | Soil texture class: clay–silt–loam Bulk density: 1.25 g cm$^{-3}$ Organic matter: 1.7% Soil pH: 8.0 FC: 0.32 m$^3$ m$^{-3}$ WP: 0.17 m$^3$ m$^{-3}$ |
| *Irrigation water characteristics* | Tagus-Segura transfer system $EC_w$: 1.3 dS m$^{-1}$ | Tagus-Segura transfer system $EC_w$: ~1.0 to 1.5 dS m$^{-1}$ |
| *Cultivation system* | The vineyards were trained to an overhead trellis system at a height of ~3.0 m above ground and covered with a high-density polyethylene mesh just above the canopy. Additionally, to prevent damage from low temperatures, rain, or hail, the vines were also covered with transparent high-density polyethylene at the end of August before harvestable picks. | |
| *Standard cultural practices* | Pruning, girdling, weed control, and phytosanitary treatments, among others, were the same for both experimental sites, and were carried out by the technical department based on the usual criteria for the area. | |
| *Climate conditions* | The climate of the study area is Mediterranean type and belongs to the Köppen "Bsh" classification, characterized by mild winters and dry and very hot summers, with an average annual temperature close to 22.5 °C, low rainfall of less than 300 mm, and a reference evapotranspiration between 1100 and 1400 mm per year [50,51]. | |

FC: field capacity, WP: wilting point, $EC_w$: irrigation water electrical conductivity.

A similar experimental design was used in both commercial vineyards. They consisted of a randomized complete block design with four block replicates per irrigation treatment. Each replicate consisted of three adjacent rows of vines, with seven vines per row. The five central vines of the central row were used for measurements, while the others served as guard vines.

The observations obtained by applying the treatments described above to the experimental units in each season and experimental site (Table 1) were grouped into a single dataset, and outliers were eliminated when they exceeded ±1.5 times the interquartile

range for each parameter evaluated, providing a total of $n = 67$ observations for firmness and $n = 64$ for the color of the berries. To determine the relationship between irrigation water reduction and yield reduction, the means of the treatments were used, and only for those where the water deficit was applied during the non-critical post-veraison period. Similarly, to determine the relationship between integral cumulative water stress during post-veraison and yield, berry firmness, and color, the variables were normalized with respect to plants without water limitations (CTL).

### 2.2. Water Stress during Post-Veraison

Vines' water status was monitored by measuring stem water potential at midday ($\Psi_s$) with a pressure chamber Model 3000 (Soil Moisture Equipment, Santa Barbara, CA, USA), which was carried out every 7–14 days from April to October, following the recommendations of Hsiao [52]. At least two shaded and mature leaves were selected in each replicate ($n = 8$ and $n = 6$ leaves per irrigation treatment for the *ES1* and *ES2*, respectively), with the leaves placed in aluminized plastic bags for at least 2 h prior to the measurements. Although vines under SDI and NI irrigation practices were subjected to water stress during the entire crop cycle, the intensity of water stress endured by each irrigation treatment was estimated with the water stress integral ($S\Psi_s$) accumulated during the post-veraison period, using the equation defined by Myers [39]:

$$S\Psi_s(\text{MPa day}) = \sum(\Psi_{i,i+1} - \Psi_c)n \tag{1}$$

where $\Psi_{i,\,i+1}$ is the mean $\Psi_s$ for any measurement $i$ and $i + 1$; $\Psi_c$ is the maximum $\Psi_s$ value measured during the post-veraison in vines without water restrictions, and $n$ is the number of days between each evaluation.

### 2.3. Yield Parameters and Irrigation Water Use Eficciency

The total yield, expressed in kilograms per vine, was determined as the sum of each harvest of the season, which began at the beginning of September and corresponded to 3–4 different dates depending on the season studied. Fruit load was determined as the number of clusters per vine and number of berries per cluster. Irrigation water use efficiency (IWUE) was determined as the ratio between yield and total irrigation applied. All these measurements were determined in all the vines used in the experiment ($n = 72$ and $n = 63$ vines per treatment for *ES1* and *ES2*, respectively). More details can be found in Conesa et al. [19].

It is important to note that to determine the effect of water stress intensity on berry yield, color, and firmness variables, these were normalized to the maximum value observed in each season and experimental site.

### 2.4. Berry Skin Color and Firmness

For berry color evaluation, the color index of red grapes (CIRG) was calculated, as it allows for an objective evaluation of the berries' external color, and more sensitively differentiates all the variations of red [53,54]. First, the CIELAB color space coordinates L* (lightness), a* (red to green), and b* (blue to yellow) were obtained by measuring three equidistant points of the equatorial zone of 15 berries per replicate ($n = 60$ berries per irrigation treatment in both experimental sites) using a Minolta CR-300 colorimeter (Minolta, Osaka, Japan) per experimental unit. From these values, the color parameters chroma (C*) and hue angle (h°) were calculated by:

$$C^* = \sqrt{(a^*)^2 + (b^*)^2}, \tag{2}$$

$$h° = \tan^{-1}(b^*/a^*) \tag{3}$$

Finally, the CIRG was determined according to the equation proposed by Carreño et al. [53] as:

$$\text{CIRG} = \left(180 - \text{h}^{\circ}\right) / \left(\text{C}^* + \text{L}^*\right) \tag{4}$$

Berry firmness, expressed as N, was evaluated in 20 randomly selected berries per replicate ($n$ = 60 berries per irrigation treatment in both experimental sites) and was obtained as the maximum force needed to break the skin by 5 mm in the equatorial zone, with a texture analyzer model LFRA 1500 (Middleboro, Brookfield, MA, USA) equipped with a 4-mm-diameter cylindrical probe moving at a speed of 10 mm s$^{-1}$.

### 2.5. Water Stress Intensity and Productive Variables' Effects on Berry Quality

A principal component analysis (PCA) was performed to explore the variability between variables and to determine the most suitable ones for predictive models of berry quality at harvest. Productive variables were used, such as total yield per vine and berry weight; fruit load, expressed as the number of clusters per vine, and number of berries per cluster; and water stress intensity, as S$\Psi_s$. The PCA was performed with the InfoStat software [55]. Since the variables had different units of measurement, the data were previously standardized, and the Pearson's r correlation matrix was used.

### 2.6. Predictive Model Algorithm

A Gaussian process (GP) is a stochastic and non-parametric method, as the model structure is determined from the data rather than through a parametric model. It defines a distribution over functions such that, if we select any two or more points in a function (i.e., different input–output pairs), observations of the outputs at these points follow a joint multivariate Gaussian distribution [42,56,57].

To predict two critical marketable parameters in colored table grape cultivars such as 'Crimson Seedless' at harvest, berry color, as the color index for red grapes (CIRG), and berry firmness (as N), based on easily determined input predictors, such as the water stress integral (S$\Psi_s$) and fruit load, two Gaussian Process Regression (GPR) algorithms were developed. The advantage of GPR over other machine learning methods lies in its seamless integration of several machine learning tasks, including hyperparameter estimation, model training, and uncertainty estimation [58], and it works well on small datasets [43].

In Gaussian Process Regression, we assume the output $y$ of an unknown function $f$ at input $x$, and it can be written as:

$$y = f(x) + \varepsilon, \tag{5}$$

where $\varepsilon$ is an independent, identically distributed Gaussian noise $\varepsilon \sim \mathcal{N}\left(0, \sigma_\varepsilon^2\right)$. GPR assumes that the function $f(x)$ is distributed as a Gaussian process:

$$f(x) \sim GP\left(m(x), k\left(x, x'\right)\right), \tag{6}$$

where $m(x)$ is the mean function and $k(x, x')$ is the covariance function or kernel. The mean function reflects the expected function value at input $x$ and the prior mean function is often set to $m(x) = 0$ to perform an inference via the covariance function. Empirically, setting the prior to 0 is often achieved by subtracting the (prior) mean from all observations. The covariance function models the dependence between the function values at different input points $x$ and $x'$. In our study, we used the radial basis function (RBF)—also called squared exponential—as the covariance function, which is the most used in GP modelling, and is defined by

$$K_{RBF}\left(x_i, x_j\right) = exp\left[-\gamma \|x_i - x_j\|^2\right], \ \gamma > 0, \tag{7}$$

where $\gamma$ is a parameter that controls the width of the Gaussian. Given a collection of inputs $X$, in regression modelling, we can instead model the transition function $f(x)$ using a GP as $f(X) \sim GP(m(X), k(X, X'))$; the vector $f = f(X)$ has a multivariate Gaussian distribution $f \mid X \sim \mathcal{N}(m(X), K)$, where $m(X)$ is the prior mean vector, and $K = k(X, X')$ is

the covariance matrix. Considering the regression model $y = f(X) + \varepsilon$, with $\varepsilon \sim \mathcal{N}(0, \sigma_\varepsilon^2)$, the distribution of the output $y$ is then $y|f \sim \mathcal{N}(f, \sigma_\varepsilon^2 \cdot I)$, and $y|x \sim \mathcal{N}(m(X), K + \sigma_\varepsilon^2 \cdot I)$, where $I$ is the identity matrix. Once a mean function and kernel are chosen, we can use the Gaussian process to draw a priori function values, as well as posterior function values conditional upon previous observations [42,56,57].

For the CIRG estimation, the dataset after outlier removal ($n = 64$) was split into training (70%) and testing (30%) sets, and a repeated k-fold cross-validation was performed with ten folds and fifty repetitions. For berry firmness estimation, the dataset ($n = 67$) was split into training (70%) and testing (30%) sets and a repeated k-fold cross-validation was performed with twenty-four folds and fifty repetitions. The metrics of model performance used were root mean squared error (RMSE), mean absolute error (MAE), and the coefficient of determination (R$^2$), given by

$$\text{RMSE} = \sqrt{\frac{\sum_{i=1}^{n}(y_i - \hat{y}_i)^2}{n}}; \tag{8}$$

$$\text{MAE} = \frac{1}{n}\sum_{i=1}^{n}|y_i - \hat{y}_i|; \tag{9}$$

$$\text{R}^2 = 1 - \frac{(y_i - \hat{y}_i)^2}{(y_i - \overline{y})^2} \tag{10}$$

where $y_i$ and $\hat{y}_i$ are the $i^{th}$ observed and predicted response for $i = 1, \ldots, n$; and $\overline{y}$ is the mean of the values.

The GPR with the RBF function was implemented using R-Studio [59] with the "caret" package [60].

## 3. Results

### 3.1. Yield Parameters and Irrigation Water Use Eficciency

The total irrigation water received by the Control (CTL) treatment, as an average of the two experimental sites, was 6754 m$^3$ ha$^{-1}$ to satisfy 100% of the reference crop evapotranspiration (ET$_0$), ranging from 5960 to 7466 m$^3$ ha$^{-1}$. The vines subjected to regulated deficit irrigation during post-veraison were irrigated with an average of 4614 m$^3$ ha$^{-1}$, corresponding to an average of 4497 and 4799 m$^3$ ha$^{-1}$ for RDI and PRD, respectively. On the other hand, those subjected to the sustained deficit irrigation (SDI) treatment during the entire crop cycle were irrigated with 3061 m$^3$ ha$^{-1}$, and those subjected to the null irrigation regime (NI) were only irrigated when the $\Psi_s$ was below $-1.2$ MPa, amounting to 1896 m$^3$ ha$^{-1}$.

Yield was negatively affected according to the water stress applied to the vines. Thus, vines not subjected to water stress showed values very close to their maximum potential. However, in those vines subjected to a water deficit, the yield was slightly reduced up to approximately a 9% reduction, in the case of RDI and PRD, with a linear decrease observed from a 30% water reduction onwards, in the case of NI (Figure 1).

In treatments where deficit irrigation was applied during the post-veraison, irrigation water use efficiency (IWUE) was described by a linear function of the accumulative water stress integral (S$\Psi_s$) in the range between 5 and 42 MPa day, increasing by 0.214 units for each MPa day of stress applied. However, when deficit irrigation was applied during critical periods, the IWUE decreased by 0.312 units, even at S$\Psi_s$ similar to post-veraison treatments, due to yield reduction, either in terms of berry quality or total yield per vine (Figure 2).

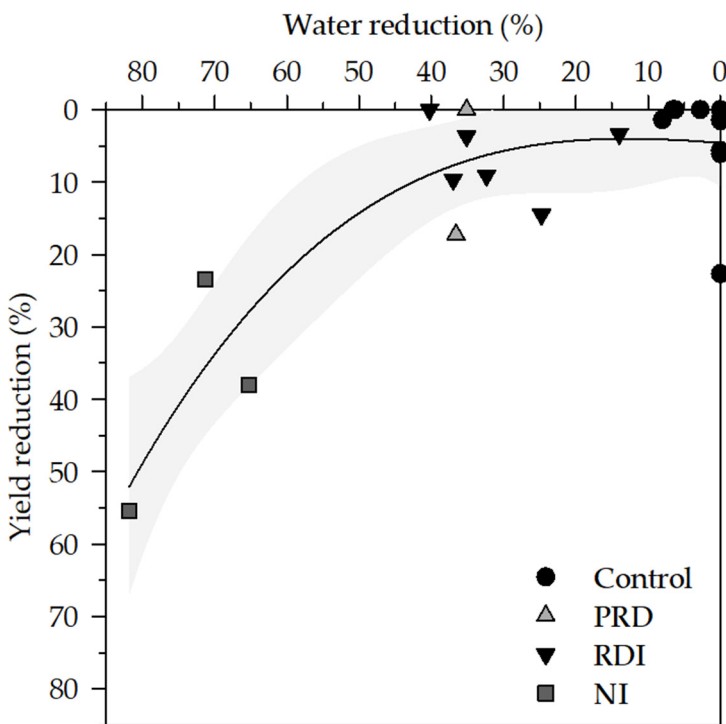

**Figure 1.** Relationship between irrigation water reduction only during the non-critical period of post-veraison and yield reduction in 'Crimson Seedless' vines under different irrigation regimes during several seasons: control (110% of $ET_c$); RDI—regulated deficit irrigation and PRD—partial root-zone drying at 50% of CTL; and NI—null irrigation, except by rainwater and supplementary irrigation when the $\Psi_s < -1.2$ MPa. Black line corresponds to regression model $y = 4.549 - 0.071x + 0.001x^2 + (8.384 \times 10^{-5})x^3$; $R^2 = 0.754$; and grey area to the 95% confidence interval. Each point corresponds to the treatment mean for each season and study site between 2011 and 2017, $n = 22$.

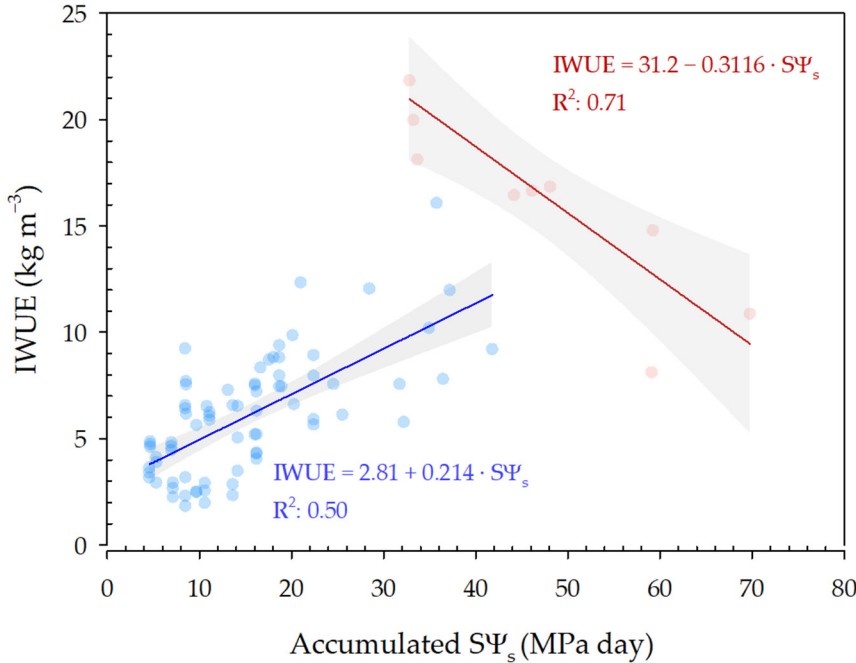

**Figure 2.** Relationship between the cumulative water stress integral ($S\Psi_s$) and the irrigation water use efficiency (IWUE) obtained in 'Crimson Seedless' vines under different irrigation regimes. Blue

circles correspond to vines without water restriction or subjected to RDI or PRD during post-veraison, without negatively affecting productive parameters. Red circles correspond to vines that were subjected to deficit irrigation during the critical periods and whose yield or berry quality was negatively affected in the short or long term (NI and SDI). $n = 67$, each point corresponds to a replicate of the irrigation treatments applied between 2011 and 2017. Lines correspond to the regression model and the grey area to the 95% confidence interval for each group of data.

### 3.2. Water Stress Intensity and Productive Variables' Effects on Berry Quality

Figure 3 shows the relationship found between water stress and yield, firmness, and berry color. The values have been expressed as normalized variables, with a value of 1 corresponding to the maximum potential of each. As water stress increased, yield and berry firmness decreased, with yield being approximately 10% more sensitive than firmness. However, berry color increased due to water stress, stabilizing the values from 30 MPa day onwards. Therefore, an optimal range of $S\Psi_s$, between 22 and 30 MPa day could be considered, since it allows the maximum productive potential (0.92) to be approached, without significantly affecting berry firmness (0.94) and increasing berry coloring (0.95) (Figure 3).

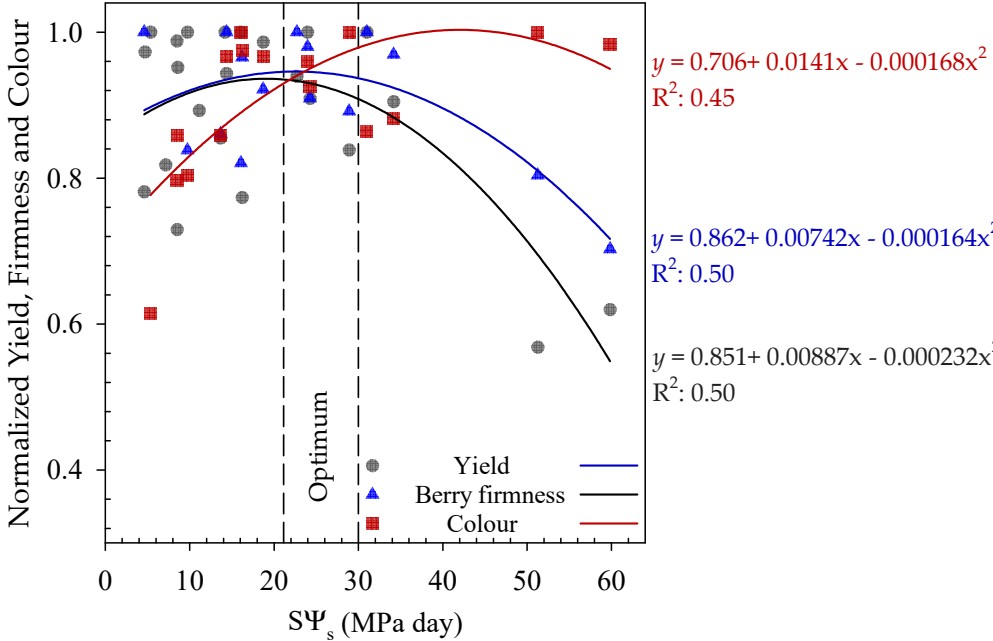

**Figure 3.** Relationship between the cumulative water stress integral during post-veraison ($S\Psi_s$) and the normalized yield (grey circles, $n = 22$), berry firmness (blue triangles, $n = 16$), and berry color (red squares, $n = 16$) with respect to 'Crimson Seedless' vines without water limitations. Each point is the mean of 3 replicates in each irrigation treatment from data obtained between 2011 and 2017. Vertical dashed lines indicate the range of optimal water stress intensity proposed for post-veraison deficit irrigation management. Continuous lines correspond to the regression models fitted for the variables.

The principal component analysis (PCA) results explained 72.0% of the total variability of the observations in its first three components (Table 1). If we consider the association coefficients between the original and transformed variables (eigenvectors), PC1 (29.6%) showed differences mainly in the productive variables: berries per cluster, cluster weight, and the number of clusters per vine. At the PC2 level, which accounted for 21.4% of the variability, the variables with the highest weight, from highest to lowest, were: $S\Psi_s$, total yield, and the CIRG. PC3 described 20.9% of the variability and was mainly based on the variation of quality variables: berry firmness and the CIRG, and the number of clusters per vine (Table 2).

**Table 2.** Principal component analysis (PCA): eigenvalues, percentage of variation accounted by the first three principal components (PC), and eigenvectors of the productive and quality variables on 'Crimson Seedless' vines under different water stress intensities during post-veraison.

|  | PC1 | PC2 | PC3 |
|---|---|---|---|
| Eigenvalue | 2.37 | 1.71 | 1.67 |
| Variance (%) | 29.6 | 21.4 | 20.9 |
| Cumulative variance (%) |  | 51.0 | 72.0 |
| SΨs | −0.11 | **0.61** | 0.08 |
| Total yield | 0.12 | **0.54** | 0.21 |
| Clusters per vine | **−0.35** | −0.07 | **0.55** |
| Berries per cluster | **0.63** | 0.02 | 0.01 |
| Cluster weight | **0.62** | 0.10 | −0.01 |
| Berry weight | −0.11 | 0.39 | 0.03 |
| Berry firmness | 0.15 | 0.12 | **0.62** |
| CIRG | −0.17 | **0.40** | **−0.52** |

Values with higher absolute weights on the determination of the PCA axes are reported in bold. $n = 64$. SΨs: cumulative water stress integral during post-veraison; CIRG: color index for red grapes.

The water stress intensity, as $SΨ_s$, was significantly correlated (Pearson's r) with berry coloring, expressed as CIRG, showing that the higher the water deficit during post-veraison, the more reddish the berry coloring. On the other hand, neither nor yield parameters were significantly correlated with berry firmness. In terms of production variables, the number of clusters per vine was significantly and negatively correlated with the cluster weight and number of berries. As expected, cluster weight was significantly correlated with the number of berries, unlike berry weight, where no significant correlation was detected (Table 3).

**Table 3.** Correlation matrix r-Pearson of the productive and quality variables on 'Crimson Seedless' vines under different water stress intensities during post-veraison.

| Variable | SΨs | | Total Yield | | Clusters per Vine | | Berries per Cluster | | Cluster Weight | | Berry Weight | | Berry Firmness | | CIRG |
|---|---|---|---|---|---|---|---|---|---|---|---|---|---|---|---|
| SΨs (MPa day) | 1.00 | | | | | | | | | | | | | | |
| Total yield (kg vine$^{-1}$) | 0.35 | ns | 1.00 | ns | | | | | | | | | | | |
| Clusters per vine | 0.20 | ns | −0.04 | ns | 1.00 | | | | | | | | | | |
| Berries per cluster | −0.05 | ns | 0.12 | ns | −0.43 | * | 1.00 | | | | | | | | |
| Cluster weight (g) | −0.03 | ns | 0.18 | ns | −0.46 | * | 0.96 | *** | 1.00 | | | | | | |
| Berry weight (g) | 0.19 | ns | 0.25 | ns | −0.02 | ns | −0.24 | ns | 0.00 | ns | 1.00 | | | | |
| Berry firmness (N) | 0.14 | ns | 0.26 | ns | 0.31 | ns | 0.20 | ns | 0.17 | ns | 0.00 | ns | 1.00 | | |
| CIRG | 0.43 | * | 0.07 | ns | −0.32 | ns | −0.21 | ns | −0.18 | ns | 0.07 | ns | −0.36 | ns | 1.00 |

SΨs: cumulative water stress integral during post-veraison; CIRG: color index for red grapes. $n = 64$. *: $p < 0.05$; ***: $p < 0.001$ and $^{ns}$: not significant.

These results show that the variables analyzed with the greatest influence on quality parameters, mainly berry color, were the intensity of water stress and fruit load, although the latter less clearly. Considering the close correlation detected between the productive variables and the opportunity to regulate the fruit load in the early season, it is possible to consider the elaboration of predictive models including the variables described above to estimate the quality of the berries at harvest.

### 3.3. Predictive Models

The dataset obtained allowed the training and validation of the predictive model on a wide range of data for the independent ($SΨ_s$ and fruit load) and dependent (berry firmness and CIRG) variables (Figure 4). In the previous section, it was observed that the $SΨ_s$ interval included values that affected berry quality and yield, so this was useful for estimating maximum reference values and optimal water stress intervals. Regarding fruit load, it varied from 30 to almost 200 clusters per vine, showing that 'Crimson Seedless'

has high fertility; therefore, it is very productive. The average berry firmness was around 10 N, and the ideal values obtained in conditions without water limitation were between the median and the upper quartile [19,21]. Regarding color, expressed as CIRG, data were grouped into the categories proposed by Carreño et al. [53], i.e., pink (2 < CIRG < 4) and red (4 < CIRG < 5) (Figure 4).

For the CIRG estimation, the coefficient of determination ($R^2$), root mean square error (RMSE), and mean absolute error (MAE) were 0.70, 0.19, and 0.16, respectively. The best $\gamma$ parameter was 0.3246 (Figure 5A).

For berry firmness estimation, the $R^2$, RMSE, and MAE were, 0.59, 1.95, and 1.64, respectively. The best $\gamma$ parameter was 0.4339 (Figure 5B).

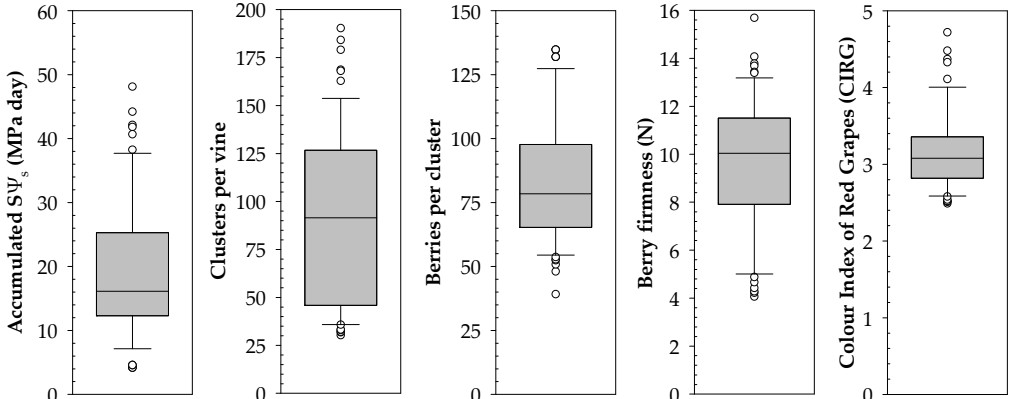

**Figure 4.** Box-and-whisker plots of the variables used to elaborate the predictive models. *n* = 67 for all the variables, except *n* = 64 for CIRG.

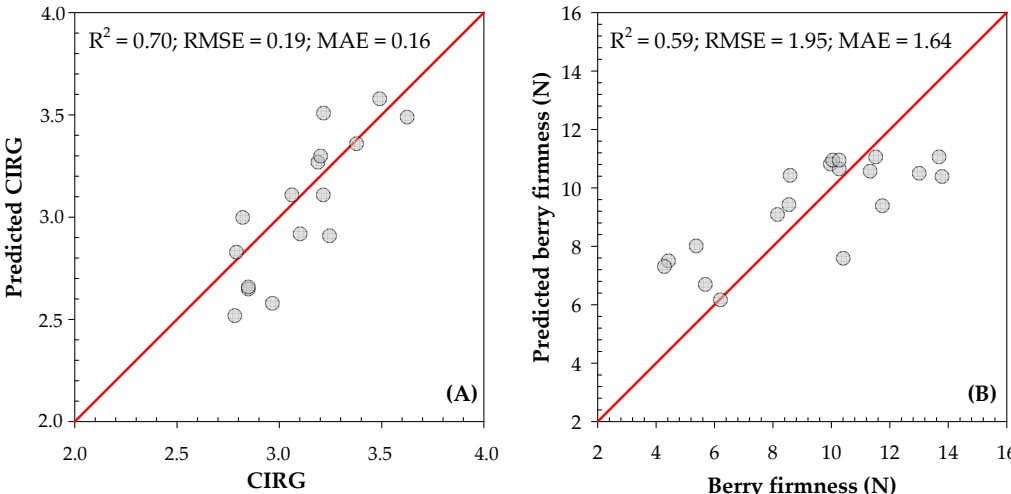

**Figure 5.** Predicted and observed color index of red grapes (CIRG) of clusters (**A**) and berry firmness (**B**) at harvest, for cv. Crimson seedless adult vines obtained through a Gaussian Process Regression with radial basis function kernel model, using as estimators the accumulated water stress integral post-veraison and fruit load, as clusters per vine and berries per cluster. The dataset for model (**A**) and (**B**) was *n* = 64 and *n* = 67, respectively.

## 4. Discussion

A water reduction of up to 40%, as compared to well-watered vines during post-veraison, together with a water stress integral of up to 30 MPa day, promotes higher color (CIRG) and firmness values in Crimson Seedless vines, without negatively affecting the final yield [4,43]. The irrigation water use efficiency (IWUE) increases linearly up to the

specified stress integral threshold value and decreases linearly beyond this threshold as yield is reduced (Figure 2).

The effect of prolonged water scarcity in semi-arid Mediterranean climates such as that found in Southeastern Spain, and the increased pressure for water resources, have highlighted the need to increase the IWUE. By means of deficit irrigation, it has been possible to increase the IWUE without negatively affecting quality or yield [4,12,61,62]. Moreover, a certain level of water stress can even improve berry quality through an increase in the red berry color and the production of health-promoting bioactive compounds [4,12,19,20], as in the case of 'Crimson Seedless' table grape production.

In this way, to carry out a successful deficit irrigation strategy, there are three fundamental factors: (i) water deficit should be carried out during periods when the crop is not sensitive, (ii) irrigation scheduling should be based on plant water status indicators, and (iii) the water stress level applied should be quantified, and the threshold reference values of $\Psi_s$ must be known. In this sense, the non-critical period for a deficit irrigation strategy in 'Crimson Seedless' is during post-veraison, with a stem water potential ($\Psi_s$) threshold of −1.2 MPa [43]. From the maximum $\Psi_s$, it is possible to quantify the water stress intensity accumulated during the established deficit period [39], allowing the extrapolation of the irrigation protocols to other agro-climatic zones.

By using these criteria, average water savings of around 40% were achieved in Crimson Seedless table grapes as compared to the water balance according to the FAO method [31], by applying either RDI or PRD irrigation strategies [62,63]. When SDI irrigation strategies were applied, the water saved was higher, even close to 70%, but it is necessary to consider that keeping the plants under deficit irrigation during the entire growth cycle can have negative effects on the production in the long term, because it causes a decrease in leaf gas exchange and vegetative growth before veraison [64].

We found that the optimum water stress intensity to achieve maximum yield potential, without affecting berry firmness and improving berry color in warm climates, corresponds to a range between 22 and 30 MPa day (Figure 3). This confirms that precise irrigation management, even with some stress to the crop, can significantly increase the water use efficiency and, in addition, reduce possible exogenous applications of agrochemicals to increase berry coloration [19,21]. Skin berry color was significantly related to the intensity of water deficit, although there was also a negative effect of fruit load on color when the vine had a high number of clusters (Table 3). These findings are consistent with the recommendation that, under conditions of high fruit load, it is necessary to thin the clusters on 'Crimson Seedless' [9]. Moreover, the combination of thinning and trunk girdling during veraison increases the concentration or modulates the pattern of anthocyanins in the skin, pigments that determine the coloring of the berries [65–67]. This cultural management allows, especially in climates with high temperatures during fruit ripening, the avoidance of coloring problems and delayed berry ripening. However, coloring is also influenced by other factors, mainly associated with the accumulation of anthocyanin pigments from veraison, regulated by a complex mechanism influenced by abscisic acid (ABA), the climatic conditions of the crop, and their variation during the season, either the maximum daytime temperature, the day/night temperature range, or the incident radiation, the soil, and the vigor of the vines [14,26,68,69]. In the same experimental site, Conesa et al. [19] also found that the accumulated ABA was associated with a decrease in the trunk growth rate at the post-veraison period. Although potential berry firmness decreases as water stress intensity is higher than 30 MPa day, no linear relationships were detected with the rest of the parameters evaluated (Table 3). In this sense, it is important to be able to predict the final berry quality in good time, so that, where and when necessary, cultural practices can be planned, such as flower or cluster thinning [5,7,9], girdling [65–67], plant growth regulator application [10,14,16–18,70,71], canopy management to improve cluster exposure to light [8,10], regulation of nitrogen fertilization [10], and deficit irrigation as a more sustainable strategy [4,12,19–21,43].

Machine learning is used to teach machines how to handle the data more efficiently when we cannot interpret the information extracted from them [72]. In supervised machine learning algorithms, a set of input variables (for instance, water stress integral and fruit load) are used to predict a response variable (for instance, berry color or firmness at harvest). The dataset is divided into training and testing sets, consisting of a set of inputs and outputs. Algorithms must "learn" patterns from the training dataset and apply them to the testing dataset for prediction or classification [73,74].

The Gaussian process (GP) is a Bayesian machine learning method that has gained attention due to its flexibility in modelling. A GP can be applied in regression analysis, called Gaussian Process Regression (GPR) [42,56]. GPRs are simple to implement, flexible, fully probabilistic models, and thus a powerful tool in many areas of application [75], even when only small datasets are available [43]. In agriculture, the use of GPR made possible the estimation, with a high goodness-of-fit, of the weekly irrigation of Crimson Seedless vines under deficit irrigation or under conditions without water limitations [43]. Moreover, the GPR method was more accurate in predicting water quality than other statistical models such as multiple linear regression or artificial neuron networks [56]. In our case, both GPR models allowed us to estimate, with a high goodness-of-fit, the berry color or firmness at harvest, defining it as a viable option when considering the effect of several water stress intensities during the non-critical period of berry development and the vines' fruit load (Figure 5).

The severe water scarcity and increasing uncertainty about the seasonal irrigation water availability faced by farmers highlight the advantage of incorporating these predictive tools, previously validated with a robust database, into agricultural decision making as a complement to the planning of cultural practices to increase IWUE and crop sustainability.

**5. Conclusions**

In 'Crimson Seedless' table grape, deficit irrigation during post-veraison with a water stress integral between 22 and 30 MPa day, and without exceeding a threshold stem water potential of $-1.2$ MPa, allowed the plant to approach its maximum productive potential, without affecting the berry firmness and increasing their coloring. It also allowed water savings of around 40% with respect to the control.

Gaussian Process Regression allows the accurate prediction of berry color and firmness at harvest, based on the water stress intensity and fruit load, which allows the consideration of cultural practices to avoid possible color problems that affect consumer acceptance.

**Author Contributions:** Conceptualization, A.P.-P. and A.T.; methodology, A.P.-P. and P.B.; software, P.B. and A.T.; validation, P.B., A.P.-P., M.R.C. and A.T.; formal analysis, P.B. and A.T.; investigation, M.R.C., A.T. and A.P.-P.; resources, P.B. and A.T.; data curation, P.B. and A.T.; writing—original draft preparation, A.P.-P. and P.B.; writing—review and editing, A.T., A.P.-P. and P.B.; visualization, P.B. and A.T.; supervision, M.R.C. and A.P.-P.; project administration, A.P.-P.; funding acquisition, A.P.-P. All authors have read and agreed to the published version of the manuscript.

**Funding:** This research was funded by the Spanish Ministry of Science of Innovation (project AGL2010-19201-C04-04) and by the National Research agency (PID2019-106226RB-C22/AEI/10.13039/501100011033), the European Union (LIFE13 ENV/ES/000539) and International Joint Programming Actions 2017 contemplated in the National R&D&I Programme oriented towards the challenges of society by the Ministry of Economy, Industry and Competitiveness—National Research Agency (AEI) (PCIN-2017-091).

**Institutional Review Board Statement:** Not applicable.

**Informed Consent Statement:** Not applicable.

**Data Availability Statement:** The data presented in this study are available on request from the corresponding author.

**Acknowledgments:** The authors thank the commercial vineyards of 'Vegafrutal' (*ES1*, belongs to the Frutas Esther S.A. company) and 'La Hornera' (*ES2*) for letting them use the facilities to carry out the study. Thanks are also due to J.M. De la Rosa, J.M. Robles, M. García-Riquelme, J.C. Ruiz-Gómez and C. Castillo, who collaborated in the research projects, for their help in field and laboratory tasks. M.R. Conesa thanks the Spanish JdlC programme (FJCI-2017-32045 and IJC2020- 045450) funded by MCIN/AEI/10.13039/501100011033 and the European Union NextGenerationEU/ PRTR.

**Conflicts of Interest:** The authors declare no conflict of interest.

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
