# Peer review of "Modelling the Impact of Water Stress during Post-Veraison on Berry Quality of Table Grapes"

_agronomy, doi:10.3390/agronomy12061416_

Round 1
Reviewer 1 Report
Review Report:
General comments:
The aim of this work was to increase the efficiency of irrigation water use in table grapes by modelling the agronomic response of the crop to different degrees of water stress during post-veraison. This work is interesting from a practical point of view to be more efficient and sustainable in irrigation in table grapes in semiarid regions under limiting water conditions and can have a great impact in the table grape growing area to save irrigation water in a high-water demand crop. The main finding of this long-term study (7 years) is that a water reduction of up to 40%, can be possible as compared to well-watered vines during post- veraison in table grapes, promoting higher color and firmness, without significant yield or quality reductions. An optimum threshold value of water stress integral of up to 30 MPa day is also proposed.
It is aceptable for publication in Agronomy journal after minor revision.
Introduction:
The introduction section is well written, clear and concise and with an adequate number of references. The problem is well explained and the objectives are clear. Perhaps there are too much self citations in introduction and results.
1. I don´t understand very well the meaning of this paragraph. Please clarify, Is it better to reduce vegetative development and to enhance berry sun exposure post-veraison or not, to improve berry quality? :
“For maximum coloring, clusters must be exposed to sunlight during ripening, so it is necessary to remove basal leaves around them and remove shoots that reduce the clusters exposure immediately following berry softening”.
Material and Methods:
2. In general, Material and Methods description is well explained and in detail.
3. Some comments:
4. I would like to know if the two vineyards were irrigated with the same water volumes for each treatment, taking into account that plant density and the number of drippers was different between the two vineyards. Please clarify.
5. I´m not familiar with these types of medelization (gaussian process regression, machine learning techniques), but I think that Model description is well explained and the results and modelization procedure can be relevant for the objectives of this study.
Results:
In general, the results are pretty clear, but I have some comments.
Figure 1 is very ilustrative and a threshold of 30-40% of wáter reductuon can be suggested by this figure. I would like to know:
how many years are used for doing this relationship in this figure 1?
Why are not used data from irrigation treatmet (SDI) in the figure 1?. How are these data?
Pag. 7-8, lines 276-281.
“Irrigation water use efficiency (IWUE) was described by a linear function of accumu lative water stress integral (SΨs ) at the post-veraison period in the range between 5 and 42 MPa day, increasing by 0.214 units for each MPa day of stress applied. However, when the SΨs was higher than 32 MPa day, IWUE decreased by 0.312 units, due to the yield reduction, either in terms of berry quality or total yield per vine (Figure 2)”
This sentence is a bit contradictory, beacuse authors say that between 5 and 42 MPa day (SΨs ) increases and at the same time authors say that SΨs was higher than 32 MPa day, IWUE decreased by 0.312 units. Please clarify this statement and threshold values.
Data from what years are used in Figure 2. How many replicates did you use to make this relationships? Please clarify in the legend.
Blue circles, red circles? Or blue lines? In figure 2.
How many years did you use to make the figure 3?
In figure 5. Observed and predicted Color Index of Red Grapes (CIRG) of clusters (A) and berry firmness (B) at harvest, for cv. Crimson seedless adult vines obtained through a Gaussian Process Regression is well correlated, which may indicate the usefulness of this modeling process used in this study (machine learning methods).
Discussion
Too long sentence and repetitive. It is not relevant for discussion. It is well known all related with Ψs. Please shorten.
Line 390-397. “Despite its limited temporal and spatial scale, Ψs has been widely validated due to its high sensitivity and the fact that as a plant measurement, it is directly related to environmental conditions and soil water availability [35–37,58]. From this indicator, it is possible to quantify the intensity of water stress by means of the cumulative water stress integral [41], allowing the extrapolation of crop irrigation protocols in 394 other agro-climatic zones, given that it is relative to the water potential of the plants without water deficit [59]. Also, the plant water status can be monitored directly or indirectly through several indicators [34,60].”
Question: From this sentence below, I would like to know if controlled DI applied during post-veraison or preveraison could reduce a high fruit load, and therefore to avoid to thin clusters in these productive vines, saving labour costs? In addition, SDI treatment could be a good option to avoid a high vigor and high fruit load in Crimson seedless?. How do SDI reduce yield compared to RDI?.
Line 410-413 “Skin berry color was significantly related to the intensity of water deficit, although there was also a negative effect of fruit load on color when the vine had a high number of clusters (Table 2). These findings are consistent with the recommendation that under conditions of high fruit load, it is necessary to thin clusters on 'Crimson Seedless' [9].”
Conclusions are clear.
Author Response
Reviewer #1:
The aim of this work was to increase the efficiency of irrigation water use in table grapes by modelling the agronomic response of the crop to different degrees of water stress during post-veraison. This work is interesting from a practical point of view to be more efficient and sustainable in irrigation in table grapes in semiarid regions under limiting water conditions and can have a great impact in the table grape growing area to save irrigation water in a high-water demand crop. The main finding of this long-term study (7 years) is that a water reduction of up to 40%, can be possible as compared to well-watered vines during post- veraison in table grapes, promoting higher color and firmness, without significant yield or quality reductions. An optimum threshold value of water stress integral of up to 30 MPa day is also proposed.
It is acceptable for publication in Agronomy journal after minor revision.
- Introduction:
The introduction section is well written, clear and concise and with an adequate number of references. The problem is well explained and the objectives are clear. Perhaps there are too much self citations in introduction and results.
Thank you for your comments and suggestion. We have modified the references and only left these to our work where strictly necessary.
- Introduction:
I don´t understand very well the meaning of this paragraph. Please clarify, Is it better to reduce vegetative development and to enhance berry sun exposure post-veraison or not, to improve berry quality? : “For maximum coloring, clusters must be exposed to sunlight during ripening, so it is necessary to remove basal leaves around them and remove shoots that reduce the clusters exposure immediately following berry softening”.
The paragraph on cultural practices was reduced. Among the canopy management, removing leaves that overshadow the cluster is an alternative to improve the colouring of the berries.
Dokoozlian, N.; Peacock, B.; Luvisi, D. Crimson Seedless Production Practices. 1989.
Dokoozlian, N.; Peacock, B.; Luvisi, D.; Vasquez, S. Cultural Practices for Crimson Seedless Table Grapes. Pub. TB 16-00 2000.
(Lines 50 – 51) “ii) Canopy management, either by regulating vines vigor or thinning leaves close to the clusters to increase its light exposure [6,9].”
- Material and Methods:
In general, Material and Methods description is well explained and in detail. Some comments: I would like to know if the two vineyards were irrigated with the same water volumes for each treatment, taking into account that plant density and the number of drippers was different between the two vineyards. Please clarify.
I´m not familiar with these types of medelization (gaussian process regression, machine learning techniques), but I think that Model description is well explained and the results and modelization procedure can be relevant for the objectives of this study.
The total irrigation water volume at both experimental sites was quite similar. Using crop coefficients adjusted for canopy cover shaded area (Williams and Ayars, 2005) reduces this effect, although, in this very vigorous cultivar, canopy cover performs similarly in less dense planting frames. The crop water requirement estimation was explained in more detail in the text.
(Lines 99 – 104) “The reference crop evapotranspiration (ET0) was obtained by the weather stations of the “Servicio de Información Agraria de Murcia” [46]. Data was computed as an average of the 7 previous days. The crop evapotranspiration (ETc) was calculated according to the FAO method (ETc = ET0 × kc) [33], with the crop coefficients (kc) reported by Williams and Ayars [47], varying between 0.2 to 0.8 according to the phenological stage.”
- Results:
In general, the results are pretty clear, but I have some comments.
Figure 1 is very ilustrative and a threshold of 30-40% of wáter reductuon can be suggested by this figure. I would like to know: how many years are used for doing this relationship in this figure 1? Why are not used data from irrigation treatmet (SDI) in the figure 1?. How are these data?.
Figure 1 is based on data collected between 2011 and 2017.
The non-critical period to water deficit previously determined and widely validated for Crimson Seedless table grapes is the post-veraison. Therefore, it was decided to consider in this relationship only those treatments where this criterion was applied. Also, sustained deficit irrigation throughout the crop cycle would affect vegetative growth and yield in the long term. The figure caption was rewritten to improve comprehension and clarify the conditions for its elaboration.
(Figure 1 caption) “Figure 1. Relationship between irrigation water reduction only during the non-critical period of post-veraison and yield reduction in ‘Crimson Seedless’ vines under different irrigation regimes during several seasons: control (110% of ETc); RDI: regulated deficit irrigation and PRD: partial root-zone drying at 50% of CTL; and NI: null irrigation, except by rainwater and supplementary irrigation when the Ψs < −1.2 MPa. Black line corresponds to regression model ; and grey area to the 95% confidence interval. Each point corresponds to the treatment mean for each season and study site between 2011 and 2017, n = 22.”
- Results:
Pag. 7-8, lines 276-281.“Irrigation water use efficiency (IWUE) was described by a linear function of accumulative water stress integral (SΨs ) at the post-veraison period in the range between 5 and 42 MPa day, increasing by 0.214 units for each MPa day of stress applied. However, when the SΨs was higher than 32 MPa day, IWUE decreased by 0.312 units, due to the yield reduction, either in terms of berry quality or total yield per vine (Figure 2)”. This sentence is a bit contradictory, beacuse authors say that between 5 and 42 MPa day (SΨs ) increases and at the same time authors say that SΨs was higher than 32 MPa day, IWUE decreased by 0.312 units. Please clarify this statement and threshold values.
Thank you for the correction. The paragraph has been rewritten, as well as the figure caption to improve comprehension.
(Lines 277 – 283) “In treatments where deficit irrigation was applied during the post-veraison, irrigation water use efficiency (IWUE) was described by a linear function of the accumulative water stress integral () in the range between 5 and 42 MPa day, increasing by 0.214 units for each MPa day of stress applied. However, when deficit irrigation was applied during critical periods, the IWUE decreased by 0.312 units, even at similar to post-veraison treatments, due to yield reduction, either in terms of berry quality or total yield per vine (Figure 2).”
- Results:
Data from what years are used in Figure 2. How many replicates did you use to make this relationships? Please clarify in the legend.
The following sentence was added “n=67 , each point corresponds to a replicate of the irrigation treatments applied between 2011 and 2017”
(Figure 2 caption) “Relationship between the cumulative water stress integral () and the irrigation water use efficiency (IWUE) obtained in 'Crimson Seedless' vines under different irrigation regimes. Blue circles correspond to vines without water restriction or subjected to RDI or PRD during post-veraison, without negatively affecting productive parameters. Red circles correspond to vines that were subjected to deficit irrigation during the critical periods and whose yield or berry quality was negatively affected in the short or long-term (NI and SDI). n = 67, each point corresponds to a replicate of the irrigation treatments applied between 2011 and 2017. Lines correspond to the regression model and the grey area to the 95% confidence interval for each group of data.”
Blue circles, red circles? Or blue lines? In figure 2.
We are sorry for this, in the manuscript uploaded on the MDPI platform in .docx all figures are correct. Probably when it was exported to .pdf it was not done correctly.
How many years did you use to make the figure 3?
The following sentence was added: Each point is the mean of 3 replicates in each irrigation treatment from data obtained between 2011 and 2017.
(Figure 3 caption) “Figure 3. Relationship between the cumulative water stress integral during post-veraison () and the normalized yield (grey circles, n = 22), berry firmness (blue triangles, n = 16) and berry color (red squares, n = 16) with respect to 'Crimson Seedless' vines without water limitations. Each point is the mean of 3 replicates in each irrigation treatment from data obtained between 2011 and 2017. Vertical dashed lines indicate the range of optimal water stress intensity proposed for post-veraison deficit irrigation management. Continuous lines correspond to the regression models fitted for the variables.”
- Discussion:
Too long sentence and repetitive. It is not relevant for discussion. It is well known all related with Ψs. Please shorten. Line 390-397. “Despite its limited temporal and spatial scale, Ψs has been widely validated due to its high sensitivity and the fact that as a plant measurement, it is directly related to environmental conditions and soil water availability [35–37,58]. From this indicator, it is possible to quantify the intensity of water stress by means of the cumulative water stress integral [41], allowing the extrapolation of crop irrigation protocols in 394 other agro-climatic zones, given that it is relative to the water potential of the plants without water deficit [59]. Also, the plant water status can be monitored directly or indirectly through several indicators [34,60].”
Done. The paragraph was rewritten.
(Lines 390 – 392) “From the maximum , it is possible to quantify the water stress intensity accumulated during the established deficit period [41], allowing the extrapolation of the irrigation protocols in other agro-climatic zones.”
- Discussion:
Question: From this sentence below, I would like to know if controlled DI applied during post-veraison or preveraison could reduce a high fruit load, and therefore to avoid to thin clusters in these productive vines, saving labour costs? In addition, SDI treatment could be a good option to avoid a high vigor and high fruit load in Crimson seedless?. How do SDI reduce yield compared to RDI?.
Line 410-413 “Skin berry color was significantly related to the intensity of water deficit, although there was also a negative effect of fruit load on color when the vine had a high number of clusters (Table 2). These findings are consistent with the recommendation that under conditions of high fruit load, it is necessary to thin clusters on 'Crimson Seedless' [9].”
As the reviewer stated a slight deficit irrigation applied especially during the pre-veraison (coinciding with full bud-break and fruit set phenological periods) might serve as a tool to reduce high fruit load, and therefore to avoid thinning clusters, promoting saving labour costs. In a previous experience (Conesa et al., 2022 https://doi.org/10.3390/w14081311), the SDI treatment reported a 65% water reduction with respect to the CTL vines but without obtained yield penalties and enhancing berry color to a higher extent than the RDI treatment (water reduction of 24% with respect to the CTL treatment). However, the NI treatment that supposed water withholding at pre-veraison (with lower evaporative demand than post-veraison), had negative impact on productive traits. Thus, it is important to know the intensity of the deficit applied during the critical period of pre-veraison
Reviewer #1:
The aim of this work was to increase the efficiency of irrigation water use in table grapes by modelling the agronomic response of the crop to different degrees of water stress during post-veraison. This work is interesting from a practical point of view to be more efficient and sustainable in irrigation in table grapes in semiarid regions under limiting water conditions and can have a great impact in the table grape growing area to save irrigation water in a high-water demand crop. The main finding of this long-term study (7 years) is that a water reduction of up to 40%, can be possible as compared to well-watered vines during post- veraison in table grapes, promoting higher color and firmness, without significant yield or quality reductions. An optimum threshold value of water stress integral of up to 30 MPa day is also proposed.
It is acceptable for publication in Agronomy journal after minor revision.
- Introduction:
The introduction section is well written, clear and concise and with an adequate number of references. The problem is well explained and the objectives are clear. Perhaps there are too much self citations in introduction and results.
Thank you for your comments and suggestion. We have modified the references and only left these to our work where strictly necessary.
- Introduction:
I don´t understand very well the meaning of this paragraph. Please clarify, Is it better to reduce vegetative development and to enhance berry sun exposure post-veraison or not, to improve berry quality? : “For maximum coloring, clusters must be exposed to sunlight during ripening, so it is necessary to remove basal leaves around them and remove shoots that reduce the clusters exposure immediately following berry softening”.
The paragraph on cultural practices was reduced. Among the canopy management, removing leaves that overshadow the cluster is an alternative to improve the colouring of the berries.
Dokoozlian, N.; Peacock, B.; Luvisi, D. Crimson Seedless Production Practices. 1989.
Dokoozlian, N.; Peacock, B.; Luvisi, D.; Vasquez, S. Cultural Practices for Crimson Seedless Table Grapes. Pub. TB 16-00 2000.
(Lines 50 – 51) “ii) Canopy management, either by regulating vines vigor or thinning leaves close to the clusters to increase its light exposure [6,9].”
- Material and Methods:
In general, Material and Methods description is well explained and in detail. Some comments: I would like to know if the two vineyards were irrigated with the same water volumes for each treatment, taking into account that plant density and the number of drippers was different between the two vineyards. Please clarify.
I´m not familiar with these types of medelization (gaussian process regression, machine learning techniques), but I think that Model description is well explained and the results and modelization procedure can be relevant for the objectives of this study.
The total irrigation water volume at both experimental sites was quite similar. Using crop coefficients adjusted for canopy cover shaded area (Williams and Ayars, 2005) reduces this effect, although, in this very vigorous cultivar, canopy cover performs similarly in less dense planting frames. The crop water requirement estimation was explained in more detail in the text.
(Lines 99 – 104) “The reference crop evapotranspiration (ET0) was obtained by the weather stations of the “Servicio de Información Agraria de Murcia” [46]. Data was computed as an average of the 7 previous days. The crop evapotranspiration (ETc) was calculated according to the FAO method (ETc = ET0 × kc) [33], with the crop coefficients (kc) reported by Williams and Ayars [47], varying between 0.2 to 0.8 according to the phenological stage.”
- Results:
In general, the results are pretty clear, but I have some comments.
Figure 1 is very ilustrative and a threshold of 30-40% of wáter reductuon can be suggested by this figure. I would like to know: how many years are used for doing this relationship in this figure 1? Why are not used data from irrigation treatmet (SDI) in the figure 1?. How are these data?.
Figure 1 is based on data collected between 2011 and 2017.
The non-critical period to water deficit previously determined and widely validated for Crimson Seedless table grapes is the post-veraison. Therefore, it was decided to consider in this relationship only those treatments where this criterion was applied. Also, sustained deficit irrigation throughout the crop cycle would affect vegetative growth and yield in the long term. The figure caption was rewritten to improve comprehension and clarify the conditions for its elaboration.
(Figure 1 caption) “Figure 1. Relationship between irrigation water reduction only during the non-critical period of post-veraison and yield reduction in ‘Crimson Seedless’ vines under different irrigation regimes during several seasons: control (110% of ETc); RDI: regulated deficit irrigation and PRD: partial root-zone drying at 50% of CTL; and NI: null irrigation, except by rainwater and supplementary irrigation when the Ψs < −1.2 MPa. Black line corresponds to regression model ; and grey area to the 95% confidence interval. Each point corresponds to the treatment mean for each season and study site between 2011 and 2017, n = 22.”
- Results:
Pag. 7-8, lines 276-281.“Irrigation water use efficiency (IWUE) was described by a linear function of accumulative water stress integral (SΨs ) at the post-veraison period in the range between 5 and 42 MPa day, increasing by 0.214 units for each MPa day of stress applied. However, when the SΨs was higher than 32 MPa day, IWUE decreased by 0.312 units, due to the yield reduction, either in terms of berry quality or total yield per vine (Figure 2)”. This sentence is a bit contradictory, beacuse authors say that between 5 and 42 MPa day (SΨs ) increases and at the same time authors say that SΨs was higher than 32 MPa day, IWUE decreased by 0.312 units. Please clarify this statement and threshold values.
Thank you for the correction. The paragraph has been rewritten, as well as the figure caption to improve comprehension.
(Lines 277 – 283) “In treatments where deficit irrigation was applied during the post-veraison, irrigation water use efficiency (IWUE) was described by a linear function of the accumulative water stress integral () in the range between 5 and 42 MPa day, increasing by 0.214 units for each MPa day of stress applied. However, when deficit irrigation was applied during critical periods, the IWUE decreased by 0.312 units, even at similar to post-veraison treatments, due to yield reduction, either in terms of berry quality or total yield per vine (Figure 2).”
- Results:
Data from what years are used in Figure 2. How many replicates did you use to make this relationships? Please clarify in the legend.
The following sentence was added “n=67 , each point corresponds to a replicate of the irrigation treatments applied between 2011 and 2017”
(Figure 2 caption) “Relationship between the cumulative water stress integral () and the irrigation water use efficiency (IWUE) obtained in 'Crimson Seedless' vines under different irrigation regimes. Blue circles correspond to vines without water restriction or subjected to RDI or PRD during post-veraison, without negatively affecting productive parameters. Red circles correspond to vines that were subjected to deficit irrigation during the critical periods and whose yield or berry quality was negatively affected in the short or long-term (NI and SDI). n = 67, each point corresponds to a replicate of the irrigation treatments applied between 2011 and 2017. Lines correspond to the regression model and the grey area to the 95% confidence interval for each group of data.”
Blue circles, red circles? Or blue lines? In figure 2.
We are sorry for this, in the manuscript uploaded on the MDPI platform in .docx all figures are correct. Probably when it was exported to .pdf it was not done correctly.
How many years did you use to make the figure 3?
The following sentence was added: Each point is the mean of 3 replicates in each irrigation treatment from data obtained between 2011 and 2017.
(Figure 3 caption) “Figure 3. Relationship between the cumulative water stress integral during post-veraison () and the normalized yield (grey circles, n = 22), berry firmness (blue triangles, n = 16) and berry color (red squares, n = 16) with respect to 'Crimson Seedless' vines without water limitations. Each point is the mean of 3 replicates in each irrigation treatment from data obtained between 2011 and 2017. Vertical dashed lines indicate the range of optimal water stress intensity proposed for post-veraison deficit irrigation management. Continuous lines correspond to the regression models fitted for the variables.”
- Discussion:
Too long sentence and repetitive. It is not relevant for discussion. It is well known all related with Ψs. Please shorten. Line 390-397. “Despite its limited temporal and spatial scale, Ψs has been widely validated due to its high sensitivity and the fact that as a plant measurement, it is directly related to environmental conditions and soil water availability [35–37,58]. From this indicator, it is possible to quantify the intensity of water stress by means of the cumulative water stress integral [41], allowing the extrapolation of crop irrigation protocols in 394 other agro-climatic zones, given that it is relative to the water potential of the plants without water deficit [59]. Also, the plant water status can be monitored directly or indirectly through several indicators [34,60].”
Done. The paragraph was rewritten.
(Lines 390 – 392) “From the maximum , it is possible to quantify the water stress intensity accumulated during the established deficit period [41], allowing the extrapolation of the irrigation protocols in other agro-climatic zones.”
- Discussion:
Question: From this sentence below, I would like to know if controlled DI applied during post-veraison or preveraison could reduce a high fruit load, and therefore to avoid to thin clusters in these productive vines, saving labour costs? In addition, SDI treatment could be a good option to avoid a high vigor and high fruit load in Crimson seedless?. How do SDI reduce yield compared to RDI?.
Line 410-413 “Skin berry color was significantly related to the intensity of water deficit, although there was also a negative effect of fruit load on color when the vine had a high number of clusters (Table 2). These findings are consistent with the recommendation that under conditions of high fruit load, it is necessary to thin clusters on 'Crimson Seedless' [9].”
As the reviewer stated a slight deficit irrigation applied especially during the pre-veraison (coinciding with full bud-break and fruit set phenological periods) might serve as a tool to reduce high fruit load, and therefore to avoid thinning clusters, promoting saving labour costs. In a previous experience (Conesa et al., 2022 https://doi.org/10.3390/w14081311), the SDI treatment reported a 65% water reduction with respect to the CTL vines but without obtained yield penalties and enhancing berry color to a higher extent than the RDI treatment (water reduction of 24% with respect to the CTL treatment). However, the NI treatment that supposed water withholding at pre-veraison (with lower evaporative demand than post-veraison), had negative impact on productive traits. Thus, it is important to know the intensity of the deficit applied during the critical period of pre-veraison
Reviewer 2 Report
Reduction of water quantities in fruit production is a hot topic in times of climate change and water scarcity.
The article focuses on the effect of reducing irrigation water in table grape production, that is a very interesting subject. It provides moreover with a predictive model that can help the farmers in decision making.
The article is well written, and all data are well presented. There are several tables and graphics, all well done. The design is appropriate, analyses are sound, and the results drawn are clearly justified. The methods are sufficiently detailed to permit replication of the study. The analyses are appropriate and well interpreted and the conclusions are justified. This is a contribution to the body of knowledge that has several novel and innovative components.
English Level is good.
Minor revision is necessary:
In Figure 1 the grey area (mentioned in description) is missing
In Figure 2 blue and red circles (mentioned in description) are missing as well as grey area
In Figure 3 the symbols (red, blue and grey, mentioned in description) are missing
Line 390-391 I would add to the sentence: “Despite its limited temporal and spatial scale, Ψs has been widely validated AS A PLANT WATER STATUS INDICATOR…”
Line 395 long sentence, not clear the last part: “…given that it is relative to the water potential of the plants without water deficit”
Line 396 if other indicators are not described, better delete this sentence.
Line 424 decrease instead of decreased
Line 427-429 this concept is widely described later in the text, so it is better to delete this sentence
Line 434-438 This sentence is better fitting at the end of the discussion chapter
Author Response
Reviewer #2:
Reduction of water quantities in fruit production is a hot topic in times of climate change and water scarcity.
The article focuses on the effect of reducing irrigation water in table grape production, that is a very interesting subject. It provides moreover with a predictive model that can help the farmers in decision making.
The article is well written, and all data are well presented. There are several tables and graphics, all well done. The design is appropriate, analyses are sound, and the results drawn are clearly justified. The methods are sufficiently detailed to permit replication of the study. The analyses are appropriate and well interpreted and the conclusions are justified. This is a contribution to the body of knowledge that has several novel and innovative components.
English Level is good.
Minor revision is necessary.
- In Figure 1 the grey area (mentioned in description) is missing
In Figure 2 blue and red circles (mentioned in description) are missing as well as grey area
In Figure 3 the symbols (red, blue and grey, mentioned in description) are missing
We are sorry for this, in the manuscript uploaded on the MDPI platform in .docx all figures are correct. Probably when it was exported to .pdf it was not done correctly.
- Line 390-391 I would add to the sentence: “Despite its limited temporal and spatial scale, Ψs has been widely validated AS A PLANT WATER STATUS INDICATOR…”
- Line 395 long sentence, not clear the last part: “…given that it is relative to the water potential of the plants without water deficit”
Done. The paragraph was rewritten. Line 396 was deleted.
(Lines 391 – 393) “From the maximum , it is possible to quantify the water stress intensity accumulated during the established deficit period [41], allowing the extrapolation of the irrigation protocols in other agro-climatic zones.”
- Line 396 if other indicators are not described, better delete this sentence.
Done. Line 396 was deleted.
- Line 424 decrease instead of decreased
Done.
(Lines 419 – 421) “In the same experimental site, Conesa et al. [20] also found that the accumulated ABA was associated with a decrease in the trunk growth rate at the post-veraison period.”
- Line 427-429 this concept is widely described later in the text, so it is better to delete this sentence
Done. The sentence was deleted.
- Line 434-438 This sentence is better fitting at the end of the discussion chapter
Done. The sentence was repositioned and rewritten to improve comprehension.
(Lines 449 – 453) “The severe water scarcity and increasing uncertainty about the seasonal irrigation water availability faced by farmers, highlight the advantage of incorporating these predictive tools, previously validated with a robust database, into agricultural decision-making as a complement to planning cultural practices to increase IWUE and crop sustainability.”
Reviewer 3 Report
Reviewer report Manuscript Number: agronomy-1733345
Title: Modelling the impact of water stress during post-veraison on berry quality of table grapes.
This document presents a study about Modelling the impact of water stress during post-veraison on the berry quality of table grapes. The text is well written and without apparent English problems. Unfortunately, the main objective is unclear and lacks a hypothesis. The authors refer to previous work of water savings in table grapes, which in some manner, reduces the novelty of this work. The work lacks coherence between the indicated in the introduction and the presented results, discussions, and conclusions. The abovementioned details and all the further observations done to the whole document oblige me to reject it as presented.
My main concerns about the document are the following:
Abstract:
Please carefully read my comments ahead, and then prepare the new abstract.
Introduction:
Lines 50-66: The information provided about the problem of the inhibition of anthocyanins, the highly detailed description of the use of PGRs, and cultural practices, could be reduced. I suggest highlighting different water management strategies and why it is crucial to model the fruit yield and quality of the grapes. Also, in this part, computer techniques such as those described in the M&M section should be briefly introduced to relate them to the presented study.
Lines 76-79: This part is critical for this work, but the presented pieces of evidence reduce its novelty. The referred works demonstrate the advantages of reducing irrigation to table grapes. Thus, the new insights about water efficiency use in this fruit crop must be highlighted. The novelty of the work must be improved.
Lines 90-93: These objectives are not agreeing with those presented in lines 17-19.
Could the authors add the central hypothesis?
Materials and methods:
Line 101: How did the authors obtain the ETc? Did they use the single or dual crop coefficients (Kc's) suggested by Allen et al., (1998) in the FAO 56 document? If the authors indicated that "…control vines were irrigated without water restrictions at 110% of ETc…", did you mean that the control treatment was over irrigated? Please provide some information about that. The following studies may provide information about the single Kc for table grapes and wine grapes that could help you to improve this part:
López-Urrea R, Montoro A, Mañas F, et al (2012) Evapotranspiration and crop coefficients from lysimeter measurements of mature 'Tempranillo' wine grapes. Agricultural Water Management 112:13–20. https://doi.org/10.1016/j.agwat.2012.05.009
Villagra P, García de Cortázar V, Ferreyra R, et al (2014) Estimation of water requirements and Kc values of Thompson Seedless table grapes grown in the overhead trellis system, using the Eddy covariance method. Chilean Journal of Agricultural Research 74:213–218. https://doi.org/10.4067/S0718-58392014000200013
Line 109: The authors obtained the stem water potential threshold of -1.2 MPa from previous experiments or literature? Please add at least one reference or more details.
Lines 204-249: There is a detailed explanation of the predictive algorithm used. This text does not agree with the indicated in the introduction section. Also, it is not clearly stated in the main objective, which lacks a clear hypothesis. The application of this predictive method to predict fruit quality and yield related to water supplied could be the main novelty of the work.
Results:
Figure 2: Where are the dots in this Figure? Please check it.
Figure 3: The same previous comment.
Lines 317-318: I'm afraid I have to disagree in discussing results that were not significant. Please focus on the significant ones. Please delete it.
Lines 352-355: This paragraph is redundant with the indicated in the M&M section. I suggest deleting it.
Figures 5 A and 5 B: They are empty figures. Please check them.
Discussions:
Lines 434-458: This part does not match with the indicated in the introduction; thus, the work lacks coherence. It has all the elements to be novel and interesting, but not as it was presented.
Conclusions:
I suggest improving this section after changing the above indicated in the document.
Author Response
Reviewer #3:
This document presents a study about Modelling the impact of water stress during post-veraison on the berry quality of table grapes. The text is well written and without apparent English problems. Unfortunately, the main objective is unclear and lacks a hypothesis. The authors refer to previous work of water savings in table grapes, which in some manner, reduces the novelty of this work. The work lacks coherence between the indicated in the introduction and the presented results, discussions, and conclusions. The abovementioned details and all the further observations done to the whole document oblige me to reject it as presented.
Thank you for the recommendations. We have reworded the manuscript, emphasising the novelty of our work. In addition, we have modified the introduction, objectives to be more in line with the results discussed.
My main concerns about the document are the following:
- Abstract
Please carefully read my comments ahead, and then prepare the new abstract.
In response to comments, the abstract has been rewritten and the objectives and novelty of the results have been highlighted.
(Lines 11 – 26) “Abstract: The aims of this work were to modelling the effect of water stress intensity during post-veraison on table grape quality and yield, as well as predicting berry quality at harvest using a machine learning algorithm. The dataset was obtained by applying different irrigation regimes in two commercial table grape vineyards during 7 growing seasons. From these data, it was possible to train and validate the predictive models over a wide range of values for the independent (water stress intensity and fruit load) and dependent (firmness and berry color) variables. The supervised learning algorithm Gaussian Process Regression allowed us to predict the variables with high accuracy. It was also determined that a reduction in irrigation of up to 40% during post-veraison, compared to vines without water limitations, and the accumulation of the water stress integral of up to 30 MPa per day, linearly increases the irrigation water use efficiency (IWUE), and promotes higher berry color and firmness. The severe water scarcity and the increasing uncertainty about the irrigation water availability for the season that farmers are facing, highlight the advantage of incorporating these validated techniques in agricultural decision making. As they allow for planning cultural practices and criteria to increase IWUE and crop sustainability”
- Introduction
Lines 50-66: The information provided about the problem of the inhibition of anthocyanins, the highly detailed description of the use of PGRs, and cultural practices, could be reduced. I suggest highlighting different water management strategies and why it is crucial to model the fruit yield and quality of the grapes. Also, in this part, computer techniques such as those described in the M&M section should be briefly introduced to relate them to the presented study.
Thanks for the suggestion. We shortened the text of the use of PGRs, and briefly described the cultural practices. We added the importance of water regime and ML experiences in agriculture.
- Introduction
Lines 76-79: This part is critical for this work, but the presented pieces of evidence reduce its novelty. The referred works demonstrate the advantages of reducing irrigation to table grapes. Thus, the new insights about water efficiency use in this fruit crop must be highlighted. The novelty of the work must be improved.
A paragraph was included to emphasize the objectives of our work and its scope. These are not related to determine the non-critical period of water deficit of Crimson Seedless, but to determine the effect of different intensities of water stress on berry quality and yield. As well as predicting the quality of the berries in the context of water scarcity by means of machine learning.
- Introduction
Lines 90-93: These objectives are not agreeing with those presented in lines 17-19.
Could the authors add the central hypothesis?
In response to comments, the objectives have been rewritten to improve comprehension.
(Lines 48 – 94) “In this sense, different cultural practices have been investigated in ‘Crimson Seedless’ to avoid these problems: i) the application of plant growth regulators such as abscisic acid and Ethephon during berry growth, but these have shown inconsistent results [10,15–19]. ii) Canopy management, either by regulating vines vigor or thinning leaves close to the clusters to increase its light exposure [6,9]. iii) Fruit load regulation, such as flowers thinning, removal of set berries or clusters thinning. This management also has the advantage that it can be carried out at the beginning of the season until the berries have reached a size of about 5 mm [6,9,10]. And, iv) the application of a deficit irrigation regime has appeared as a more sustainable alternative to prevent berry coloring problems and also promote the production of bioactive compounds [4,20,21].
Deficit irrigation (DI) can increase berry color and cluster homogeneity at harvest, and at the same time, increase the irrigation water use efficiency (IWUE) without negatively affecting yield or berry quality [4,12,20,22,23]. The effect of climate change has increased the intensity of water scarcity in Mediterranean areas, so different DI strategies have been studied instead [24,25]. The most common methods are regulated deficit irrigation (RDI) [26] and partial root-zone drying (PRD) [27]. Both provide less irrigation during periods of the crop that are not sensitive to water deficit. In ‘Crimson Seedless’, the non-critical period is during post-veraison [12,22]. Another DI method that can increase the color of grape berries is sustained deficit irrigation (SDI), although in contrast to RDI, the irrigation reduction is applied during the entire crop cycle [28–30]. However, the effect of SDI would reduce crop yield and vegetative growth in the long-term [31,32]. Generally, the reduction in irrigation is estimated from the FAO water balance [33], but it is necessary to complement it with a method to control the plant water status [34]. In this sense, the most validated plant water status indicator is the stem water potential (Ψs), as it is directly related to environmental conditions and soil water availability [35–40]. Furthermore, the water stress integral [41] is an appropriate tool for quantifying the water stress applied as a function of the Ψs, and to extrapolate the protocols obtained to other agro-climatic zones. Therefore, determining the magnitude of the optimal crop water stress is essential to increase the sustainability of the production.
The irrigation volume available to the farmer is increasingly uncertain, so it is necessary to explore the relationships between the intensity of the water stress to be applied and its effect on berry quality, as well as its interaction with cultural practices to optimize the production. The incorporation of machine learning algorithms has made it possible to obtain models for several uses in agriculture [42,43]. Thus, using a Gaussian Process Regression [44], the weekly water requirement of 'Crimson Seedless' vines could be estimated with high accuracy from daily maximum temperature and day of the year [45]. To obtain robust models, it is necessary to have a reliable database, obtained under controlled experimental conditions and with a wide range of values to train and validate the model in different scenarios. With this premise, our research was carried out using data obtained during several seasons from the research of our team in two experimental sites and with vines subjected to a wide range of water stress.
Therefore, our research aims to (i) modelling the effect of water stress intensity on berry quality and yield, and (ii) predict berry quality at harvest in a Mediterranean climate with severe water scarcity, based on two easily applicable and quantifiable parameters: water stress integral and fruit load. Both objectives were developed to provide farmers with a tool to cope with water scarcity while maintaining the production sustainability and increasing the irrigation water use efficiency.”
- Materials and methods
Line 101: How did the authors obtain the ETc? Did they use the single or dual crop coefficients (Kc's) suggested by Allen et al., (1998) in the FAO 56 document? If the authors indicated that "…control vines were irrigated without water restrictions at 110% of ETc…", did you mean that the control treatment was over irrigated? Please provide some information about that. The following studies may provide information about the single Kc for table grapes and wine grapes that could help you to improve this part:
López-Urrea R, Montoro A, Mañas F, et al (2012) Evapotranspiration and crop coefficients from lysimeter measurements of mature 'Tempranillo' wine grapes. Agricultural Water Management 112:13–20. https://doi.org/10.1016/j.agwat.2012.05.009
Villagra P, García de Cortázar V, Ferreyra R, et al (2014) Estimation of water requirements and Kc values of Thompson Seedless table grapes grown in the overhead trellis system, using the Eddy covariance method. Chilean Journal of Agricultural Research 74:213–218. https://doi.org/10.4067/S0718-58392014000200013
Thank you for the recommendation. The method for determining the crop water requirement and the crop coefficients used have been incorporated. In the irrigation scheduling, the climatic data used corresponded with the previous week. For this reason, the percentage of ETc was increased to 110% to satisfy the crop water needs during the summer hot and dry days that might be occurred and thus it would affect the CTL treatment. This fact does not affect the conclusions, because the deficit treatments were applied with respect to the CTL. For a better understating, the text was rewritten. Also, the word "to avoid" was added to the sentence " vines were irrigated at 110% of the ETc to avoid water restrictions throughout the irrigation season ".
(Lines 100 – 105) “The reference crop evapotranspiration (ET0) was obtained by the weather stations of the “Servicio de Información Agraria de Murcia” [46]. Data was computed as an average of the 7 previous days. The crop evapotranspiration (ETc) was calculated according to the FAO method (ETc = ET0 × kc) [33], with the crop coefficients (kc) reported by Williams and Ayars [47], varying between 0.2 to 0.8 according to the phenological stage”.
- Materials and methods
Line 109: The authors obtained the stem water potential threshold of -1.2 MPa from previous experiments or literature? Please add at least one reference or more details.
Thank you for the recommendation. The reference to the threshold stem water potential for the same cultivar has been incorporated.
(Lines 114 – 116) “…when the stem water potential (Ψs) was below the threshold of −1.2 MPa previously determined for ‘Crimson Seedless’ [51].”
- Materials and methods
Lines 204-249: There is a detailed explanation of the predictive algorithm used. This text does not agree with the indicated in the introduction section. Also, it is not clearly stated in the main objective, which lacks a clear hypothesis. The application of this predictive method to predict fruit quality and yield related to water supplied could be the main novelty of the work.
In response to the comments, the introduction has been reformulated to clearly highlight the objectives and novelty of the research.
- Results
Figure 2: Where are the dots in this Figure? Please check it.
Figure 3: The same previous comment.
We are sorry for this, in the manuscript uploaded on the MDPI platform in .docx all figures are correct. Probably when it was exported to .pdf it was not done correctly.
- Results
Lines 317-318: I'm afraid I have to disagree in discussing results that were not significant. Please focus on the significant ones. Please delete it..
Done. The sentence was deleted.
- Results
Lines 352-355: This paragraph is redundant with the indicated in the M&M section. I suggest deleting it.
Done. The paragraph was deleted.
- Results
Figures 5 A and 5 B: They are empty figures. Please check them.
We are sorry for this, in the manuscript uploaded on the MDPI platform in .docx all figures are correct. Probably when it was exported to .pdf it was not done correctly.
- Discussions
Lines 434-458: This part does not match with the indicated in the introduction; thus, the work lacks coherence. It has all the elements to be novel and interesting, but not as it was presented.
In response to comments, the abstract has been rewritten and the objectives and novelty of the results have been highlighted, so that the introduction is consistent with what is described in the results-discussion and finally, in the conclusions.
(Lines 401 – 453) “We found that the optimum water stress intensity to achieve maximum yield potential, without affecting berry firmness and improving berry color in warm climates, corresponds to a range between 22 and 30 MPa day (Figure 3). This confirms that precise irrigation management, even with some stress to the crop, can significantly increase the water use efficiency and, in addition, reduce possible exogenous applications of agrochemicals to increase berry coloration [20,22]. Skin berry color was significantly related to the intensity of water deficit, although there was also a negative effect of fruit load on color when the vine had a high number of clusters (Table 2). These findings are consistent with the recommendation that under conditions of high fruit load, it is necessary to thin clusters on 'Crimson Seedless' [9]. Also, the combination of thinning and trunk girdling during veraison increases the concentration or modulates the pattern of anthocyanins in the skin, pigments that determine the coloring of the berries. [67–69]. This cultural management allows, especially in climates with high temperatures during fruit ripening, the avoidance of coloring problems and delayed berry ripening. However, coloring is also influenced by other factors, mainly associated with the accumulation of anthocyanin pigments from veraison, regulated by a complex mechanism influenced by abscisic acid (ABA), the climatic conditions of the crop, and their variation during the season, either the maximum daytime temperature, the day/night temperature range or the incident radiation, the soil, and the vigor of the vines. [14,15,28,70,71]. In the same experimental site, Conesa et al. [20] also found that the accumulated ABA was associated with a decrease in the trunk growth rate at the post-veraison period. Although potential berry firmness decreases as water stress intensity is higher than 30 MPa day, no linear relationships were detected with the rest of the parameters evaluated (Table 2). In this sense, it is important to be able to predict the final berry quality in good timing, so that, where and when necessary, cultural practices can be planned, such as flower or cluster thinning [5,6,9], girdling [67–69], plant growth regulators application [10,15,16,19,72–74], canopy management to improve cluster exposure to light [8,10], regulation of nitrogen fertilization [10], and deficit irrigation as a more sustainable strategy [4,12,20–22,45].
Machine learning is used to teach machines how to handle the data more efficiently when we cannot interpret the information extracted from it [75]. In supervised machine learning algorithms, a set of input variables (for instance, water stress integral and fruit load) are used to predict a response variable (for instance, berry color or firmness at harvest). The dataset is divided into training and testing sets, consisting of a set of inputs and outputs. Algorithms must “learn” patterns from the training dataset and apply them to the testing dataset for prediction or classification [76,77].
The Gaussian process (GP) is a Bayesian machine-learning method which has gained attention due to its flexibility in modeling. A GP can be applied in regression analysis, called Gaussian process regression (GPR) [58,60]. GPR are simple to implement, flexible, fully probabilistic models, and thus a powerful tool in many areas of application [78] even when only small datasets are available [45]. In agriculture, the use of GPR made possible the estimation, with a high goodness-of-fit, of the weekly irrigation of Crimson Seedless vines under deficit irrigation or under conditions without water limitations [45]. Also, the GPR method was more accurate in predicting water quality than other statistical models such as multiple linear regression or artificial neuron networks [58]. In our case, both GPR models allowed us to estimate, with a high goodness-of-fit, the berry color or firmness at harvest, defining it as a viable option when considering the effect of several water stress intensities during the non-critical period of berry development and the vines fruit load (Figure 5).
The severe water scarcity and increasing uncertainty about the seasonal irrigation water availability faced by farmers, highlight the advantage of incorporating these predictive tools, previously validated with a robust database, into agricultural decision-making as a complement to planning cultural practices to increase IWUE and crop sustainability”
- Conclusions
I suggest improving this section after changing the above indicated in the document.
In response to comments, the abstract has been rewritten and the objectives and novelty of the results have been highlighted, so that the introduction is consistent with what is described in the results-discussion and finally, in the conclusions.
Round 2
Reviewer 3 Report
All of the suggested observations were fulfilled. I have no new observations. Congrats.